# Novel Nanosized Spinel MnCoFeO_4_ for Low-Temperature Hydrocarbon Oxidation

**DOI:** 10.3390/nano12213900

**Published:** 2022-11-04

**Authors:** Vencislav Tumbalev, Daniela Kovacheva, Ivanka Spassova, Ralitsa Velinova, Georgi Tyuliev, Nikolay Velinov, Anton Naydenov

**Affiliations:** 1Institute of General and Inorganic Chemistry, Bulgarian Academy of Sciences, 1113 Sofia, Bulgaria; 2Institute of Catalysis, Bulgarian Academy of Sciences, 1113 Sofia, Bulgaria

**Keywords:** MnCoFeO_4_, spinel, solution-combustion synthesis, complete oxidation, hydrocarbons

## Abstract

The present paper reports on MnCoFeO_4_ spinels with peculiar composition and their catalytic behavior in the reactions of complete oxidation of hydrocarbons. The samples were synthesized by solution combustion method with sucrose and citric acid as fuels. All samples were characterized by powder X-ray diffraction, N_2_-physisorption, scanning electron microscopy, thermal analysis, X-ray photoelectron spectroscopy, and Mössbauer spectroscopy. The catalytic properties of the spinels with Mn:Co:Fe = 1:1:1 composition were studied in reactions of complete oxidation of methane, propane, butane, and propane in the presence of water as model pollutants. Both prepared catalysts are nanosized materials. The slight difference in the compositions, structure, and morphology is due to the type of fuel used in the synthesis reaction. The spinel, prepared with sucrose, shows a higher specific surface area, pore volume, higher amount of small particles fraction, higher thermal stability, and as a result, more exposed active sites on the sample surface that lead to higher catalytic activity in the studied oxidation reactions. After the catalytic tests, both samples do not undergo any substantial phase and morphological changes; thus, they could be applied in low-temperature hydrocarbon oxidation reactions.

## 1. Introduction

Air pollution control is one of the main concerns of humanity nowadays. Volatile organic compounds (VOCs) are recognized as one of the major classes of pollutants. The recent low permissible values for them are imposed by the fact that VOCs have a harmful effect on living organisms. VOC emissions in the atmosphere participate in the destruction of stratospheric ozone and in the formation of photochemical smog, and some of them are the cause of the greenhouse effect. The sources of these harmful gas emissions could be diverse: chemical and petrochemical enterprises, oil and natural gas processing processes, energy production, machine industry, printing, food production, etc. [1]. The neutralization of VOCs can be achieved by different methods, where the catalytic one is the most effective [2]. Catalysts based on supported noble metals exhibit the highest efficiency in such processes [3,4,5]. However, their high cost, limited abundance, and deactivation at high temperatures make their wide use unprofitable. On the other hand, supported noble metals catalysts deteriorate in the presence of catalytic poisons [6]. These disadvantages encourage efforts to find a good alternative for their substitution with new materials. The transition metal oxides can be regarded as very prospective for the complete oxidation of VOCs application. Among them, various oxides of Cu, Mn, Co, Fe, Cr, and Ni are the most active [7,8,9,10]. Special interest is paid to binary oxides of the mentioned transition metals (in particular, those with a spinel structure). The general formula of compounds crystallizing in this structural type is AB_2_X_4_, where A and B are cations of appropriate oxidation state, and the anion X can be O, Se, S, Te, etc. The structure of most spinel compounds is described in the space group Fd-3m. The anions are arranged in a three-layer cubic dense packing. The cations are stuffed in this packing, partially occupying 1/2 of the octahedral and 1/8 of the available tetrahedral interstices. For spinel oxides, A is usually a metal ion in the oxidation state of 2+, and B is a metal ion in the 3+ oxidation state. In a normal spinel structure, the A ions occupy the tetrahedral sites, and the B ions occupy the octahedral sites—(A)[B_2_]O_4_. The cation distribution of the inverse spinel allows the 3+ metal ion to occupy the tetrahedral positions, while 2+ cations and the rest of the 3+ cations occupy the octahedral positions—(B)[AB]O_4_. Intermediate cation distributions are considered partially inverse. The spinel structure favors high resistivity to catalytic poisons because of the presence of structural defects, electron exchange between neighboring ions, and the possibility for surface recovery. Many different spinel compositions have been proposed for catalytic VOC oxidation—ferrites, cobaltites, manganites, chromites, etc. [11,12,13,14,15]. The spinel ferrites offer high activity and stability during this reaction due to their high specific surface area and small particle sizes [16,17]. It is worth mentioning that the production of spinel ferrites is cheap due to the availability of raw materials, thus motivating investigations on ferrite catalysts.

The catalytic oxidation of hydrocarbons not only depends on the particle size of the active oxide catalyst but is also structure sensitive. It is considered that the metal ions occupying the octahedral sites in the spinel structure are more predominantly responsible for the catalytic activity of materials with spinel structure than those occupying the tetrahedral sites [18,19,20]. The different coordination influences the metal orbital energies and spatial orientation, which leads to a different interaction with oxygen species [21,22]. On the other hand, in spinel oxide ferrites, cations in octahedral sites are mainly exposed to the surface; hence, the catalytic activity of these materials is closely related to the type of cations in the octahedral sites. For these reasons, the normal spinel structure favors the exposition of only Fe^3+^ ions, the latter having relatively low catalytic activity. For example, ZnFe_2_O_4_ shows normal cation distribution, NiFe_2_O_4_ is a representative of inverse spinels, while MnFe_2_O_4_ and CoFe_2_O_4_ are partially inverse spinels and the degree of the inversion depends on the preparation conditions. Our previous paper it is reported the influence of the type of fuel used in solution combustion synthesis on the catalytic properties in VOCs oxidation of MnFe_2_O_4_ [23]. It was found that specific structural and morphological characteristics of the prepared spinels are responsible for their oxidation properties at relatively low temperatures. In order to increase the catalytic activity in the complete oxidation of hydrocarbons, an attempt for cation compositional variation is appropriate by adding a third cation into the spinel structure. The selected stoichiometry for the present study is MnCoFeO_4_. The literature data for this triple spinel are scarce and contradictory, even though some of them do not report the same stoichiometry [24,25,26,27]. The oxidation state of manganese, cobalt, and iron cations, in this case, is unclear and may depend on the synthesis conditions, thus raising various cation distributions and interactions. Some authors [26] reported that oxidation states of Mn, Co, and Fe are (+II), (+III), and (+III), respectively, and a normal spinel is realized. In contrast, in [27], the ions found are Mn^3+^ and Co^2+^, and partially inverse spinel is observed [28].

The aim of the present paper is the synthesis of spinels with composition MnCoFeO_4_, where Mn:Co:Fe = 1:1:1 by the solution combustion method and the study of their catalytic behavior in the reaction of complete oxidation of n-alkanes—methane, propane, and butane. The novelty of the present work is in the originality of the composition and the synthesis method, allowing nanosized materials with prospective catalytic properties.

## 2. Materials and Methods

### 2.1. Sample Preparation

Finely powdered MnCoFeO_4_ materials were synthesized via the solution combustion method. Starting reagents and oxidizers in the synthesis reaction were Mn(NO_3_)_2_.4H_2_O (Sigma-Aldrich, Saint Louis, MO, USA), Fe(NO_3_)_3_.9H_2_O (Sigma-Aldrich, Saint Louis, MO, USA), and Co(NO_3_)_2_.4H_2_O (Sigma-Aldrich, Saint Louis, MO, USA). Metal nitrates were taken in a molar ratio of 1:1:1. As fuel and reducing agents in the combustion reaction, sucrose—C_12_H_22_O_11_ (Valerus, Sofia, Bulgaria) and citric acid—C_6_H_8_O_7_ (Valerus, Sofia, Bulgaria) were used. The amounts of fuel and the oxidizers were taken to obey the 1:1 ratio, according to the principles of combustion described by Jain et al. [29]. The combined metal nitrates and the fuel water solutions were mixed and slowly evaporated on a hot plate. After the removal of water, a spontaneous flame reaction between the oxidizer and the reducer takes place. The reaction is accompanied by the evolvement of a large number of gases. With the used fuel–citric acid and sucrose, the visual observation shows a more intensive reaction for citric acid. The difference is that sucrose shows a transformation to a voluminous foam-like mass before the ignition. This specific feature is a chain of reactions known as caramelization, and the combustion reaction with sucrose is mild with probably lower local temperature. Powder residue containing mixed metal oxides was obtained after several minutes. The final products were thermally treated at 400 °C for 1 h and at 500 °C for 1 h. The samples were denoted as MCF-S (sucrose fuel) and MCF-CA (citric acid fuel).

### 2.2. Sample Characterization

Powder X-ray Diffraction (PXRD) patterns were collected in the range of 5–90° 2θ on a Bruker D8-Advance Diffractometer equipped with Cu tube (λ = 1.5418 Å) and LynxEye detector. The evaluation of the diffraction patterns was performed using the EVA software package in combination with the ICDD-PDF-2 database. Structure refinements were made using Topas-4.2 program (Karlsruhe, Germany). The mean coherent domain sizes were evaluated from diffraction peak broadening. The diffraction peaks approximation was performed by means of the fundamental parameters approach implemented in the program [30].

The morphology of the obtained mixed oxide materials was investigated using a JEOL-JSM-6390 scanning electron microscope. The determination of the elemental composition was performed by energy dispersive spectroscopy (EDS) on the same equipment.

The texture of mixed oxides (specific surface area, pore volume, and pore size distribution) was determined by low-temperature nitrogen adsorption at −196 °C using a Quantachrome Nova 1200e instrument (Boynton Beach, FL, USA). The specific surface area was calculated using the Brunauer, Emmett, and Teller (BET) equation, and the total pore volume and average pore diameter were determined according to the Gurvich rule at p/p_0_ ≈ 0.99. The pore-size distributions were determined by the Barrett–Joyner–Halenda (BJH) method using the desorption branch of the isotherm.

The thermal analysis (TG/DTA) was made in a LABSYSEVO-1600 (SETARAM, Caluire-et-Cuire, France) in a synthetic air stream in the temperature range 25–800 °C and heating rate of 10 K/min.

The XPS measurements were carried out in the electron spectrometer Escalab-MkII (VG Scientific, Manchester, UK)) with a pressure of ~5 × 10^−8^ Pa. The O1s, Mn2p, Fe2p, and Co2p photoelectron lines were recorded. Surface composition was determined by using the normalized photoelectron intensities [31].

The Mössbauer measurements were performed with a Wissel (Wissenschaftliche Elektronik GmbH, Starnberg, Germany) electromechanical spectrometer working in a constant acceleration mode at room temperature (RT) and at liquid nitrogen temperature (LNT). A ^57^Co/Rh source (Activity ≅ 20 mCi) and an α-Fe standard were used. The experimentally obtained spectra were fitted with WinNormos. The parameters of hyperfine interaction, such as isomer shift (δ), quadrupole splitting (Δ), quadrupole shift (2ε), effective internal magnetic field (B_hf_), line widths (Γ_exp_), and relative weight (G) of the partial components in the spectra were determined.

### 2.3. Catalytic Tests

The tests on complete catalytic oxidation of different hydrocarbons on MnCoFeO_4_ catalysts were carried out in a fixed bed reactor with a gaseous hourly space velocity (GHSV_STP_) of 100,000 h^−1^. The catalytic activity measurements were performed at the following inlet concentrations of the hydrocarbons: 990 ppm methane; 330 ppm propane; 245 ppm butane. The oxygen concentration was 16 vol.%.

For estimation of kinetics parameters for the propane combustion, the concentrations of the reactants were varied as follows: popane levels at 170, 330, 500, and 670 ppm; oxygen levels at 1.5, 3, 8, and 16 vol.%; water vapor at levels 0 and 2 vol.%. The residual squared sum (RSS) between the experimental data and the model predictions was minimized, and the square of the correlation coefficient (R^2^) was calculated.

All gaseous mixtures were balanced to 100 vol.% with nitrogen (4.0). The water vapor during propane tests was added at a concentration of 2 vol.%. All gasses were supplied by Messer, Sofia, Bulgaria. The total catalyst bed volume was 0.5 cm^3^ (0.3 cm^3^ catalyst and 0.2 cm^3^ quartz–glass particles), and the reactor diameter was 6 mm (D_reactor_/D_particles_ ≥ 10). On-line gas analyzers (CO/CO_2_/O_2_, Teledyne) and THC-FID (Thermo FID-TG, SK Elektronik GmbH, Leverkusen, Germany) for analysis of total hydrocarbon content, equipped with a flame ionization detector, were applied for the converted gas mixture analysis.

## 3. Results

### 3.1. X-ray Diffraction

The XRD patterns of the fresh and used catalysts are presented in Figure 1. All patterns can be indexed in cubic space group Fd-3m, typical for the spinel structure. Peaks of individual oxides or other impurity phases were not detected. However, a closer look reveals a high angle asymmetry of all peaks, which cannot be attributed to instrumental effect since the XRD of the samples obtained after thermal analyses up to 800 °C (Appendix A) do not show any asymmetry. The Rietveld refinement of the crystal structure of the MCF-S and MCF-CA was first attempted with a single cubic spinel phase. The results are presented in Appendix A, respectively. The difference plot and the values for the reliability factors clearly indicate a bad fit. The observed asymmetry cannot be interpreted with tetragonal cells with lower symmetry of the spinel structure. Another possibility for interpretation of this observation is to assume the simultaneous synthesis of two spinels with close unit cell parameters and different crystallite sizes. Applying this hypothesis, we obtain a good pattern fit for the fresh samples with a combination of two spinels—Appendix A. In both figures, the set of lines corresponding to one of the phases has a narrow width, and according to Scherrer’s relation, this phase has a larger crystallite size. The second phase, with broader lines, has a smaller crystallite size, respectively. The two spinels have mean crystallite sizes of 15 nm and 5 nm in quantities of 25 mass.%: 75 mass.% for the MnCoFeO_4_ obtained using sucrose as a fuel. For the material obtained with the use of citric acid, the mean crystallite sizes of the two spinels are 22 nm and 5 nm, and their ratio is 40 mass.%: 60 mass.%. After the catalytic tests, these ratios for both samples became 45 mass.%: 55 mass.%, implying some increase in the amount of larger crystallites. The triple cation set gives possibilities for a variety of spinel compositions, such as CoFe_2_O_4_, MnFe_2_O_4_, CoMn_2_O_4_, FeMn_2_O_4_, etc., with various cation distributions. A possible explanation for the formation of two spinels in both cases is that the use of different fuels results in a combustion reaction with different parameters (ignition temperature, reaction rate, the maximal temperature reached, etc.). In the present case, these parameters and the subsequent thermal treatment somehow favor the synthesis of two spinel phases with closer enthalpies of formation. The thermal treatment of the samples at higher temperatures led to spinel phase decomposition (Appendix A).

### 3.2. Nitrogen Adsorption

The texture characteristics of the obtained samples were examined by measurements of their N_2_ adsorption–desorption isotherms, shown in Figure 2 and in Table 1.

The isotherm of MCF-S is of Type IV with a hysteresis loop of Type H1, according to the IUPAC classification [32,33]. This type of isotherm reveals mesoporous material, and the type H1 hysteresis is mostly found in materials with uniform mesopores. The isotherm of MCF-CA is of complex Type II and Type IV (pseudo-type II) with an H3 hysteresis loop. The isotherm evidenced macro-mesoporous material, while such a hysteresis loop is characteristic of non-rigid aggregates of plate-like particles. Pore-size distributions evidenced the formation of nanosized material in both preparation procedures.

One can see that the sample prepared with fuel sucrose (MCF-S) possesses a larger specific surface area, total pore volume, and slightly bigger average pore diameter than this, prepared with citric acid (MCF-CA) as a fuel. Depending on the morphology of the particles, the pores that are formed between them have a very complex shape. In the present case, two types of voids are formed between the differently shaped particles giving apparent dual porosity (Figure 2).

### 3.3. Scanning Electron Microscopy

Figure 3 presents the SEM images for both solution combustion prepared triple cation spinels.

One could observe that the fuel used influences the morphology of the prepared samples. It is seen from Figure 3a,b that the ferrite sample prepared with sucrose as a fuel (MCF-S) is more dispersed, having clusters formed by particles with similar sizes and shapes. The ferrite sample, prepared with citric acid as a fuel (MCF-CA), has more rough morphology (Figure 3c,d). In it, two types of particles formed could be clearly seen—with small sizes and an almost round shape and plate-like particles with larger sizes. The results from the SEM analysis confirm general findings from XRD (two types of crystallites) and nitrogen physisorption analyses (two types of pores).

### 3.4. Thermal Analysis

The thermal stability of the formed spinels was investigated by combined thermal analyses (TG-DTA). The results for both spinels MCF-S and MCF-CA are shown in Figure 4. It can be seen that the thermal transformation occurs in two stages. The first mass loss stage ends at about 200 °C, which can be attributed to dehydration for both samples. The second stage ended at about 750 °C and could be associated with the rearrangement and degradation of the formed spinels. The temperatures of the thermal effects coincide for both spinels, but the degree of mass loss is different. In the case of MCF-S, the mass change is about 7%, while for MCF-CA, the mass loss is 4.5%. Small exothermal effects at 205 °C and 315 °C in the DTA curves are attributed to the spinel rearrangement. The pronounced exoeffect at about 650 °C could be due to either sintering [34] or decomposition and rearrangement in the crystal structure of the spinel phases as a result of partial reduction of the cations [35]—see Appendix A.

### 3.5. Mössbauer Study

Mössbauer spectra of both samples are shown in Figure 5. The spectra at room temperature (Figure 5a) are composed of a combination of sextet and doublet components.

A model containing two sextets and one doublet was used for the mathematical processing of the spectra. Sextet parameters corresponding to Fe^3+^ in a tetrahedral coordination (with lower isomer shift) and Fe^3+^ in an octahedral coordination (with higher isomer shift) in a spinel-type structure were obtained. The doublet components are probably due to small ferrite moieties that exhibit superparamagnetic properties. The relative weight of the doublet (Table 2) in MCF-S (G = 65%) is higher than that of the doublet in MCF-CA (G = 37%); hence, MCF-S is more finely crystalline than MCF-CA. These results are in full agreement with those obtained by XRD, SEM, and nitrogen adsorption analyses.

In order to determine more precisely the nature of the doublets as well as the distribution of iron in the spinel structure, measurements were made at liquid nitrogen temperature (LNT = −196 °C). Almost complete disappearance of the doublets can be seen from the spectra at LNT (Figure 5b), which proves the assumption of the presence of particles with superparamagnetic behavior. Unfortunately, the temperature of liquid nitrogen is not low enough to prevent relaxation phenomena—a sextet of broad lines (Sx3) is present in both spectra. This makes it difficult to accurately determine the distribution of iron in tetrahedral and octahedral positions. The higher values of the internal magnetic field (B_hf_) prove the presence of larger crystallites in the sample synthesized with citric acid compared to the one synthesized with sucrose.

### 3.6. X-ray Photoelectron Spectroscopy

X-ray photoelectron spectroscopy is a useful analysis for the identification of the oxidation states of the constituents as well as their content on the surface. Figure 6 presents the XPS spectra of O1s, Mn2p, Fe2p, and Co2p of MCF-S and MCF-CA.

The oxygen O1s line shows asymmetry towards the higher binding energies and could be decovoluted into three peaks at 530.3 eV, 531.9 eV, and 533.4 eV, evidencing several metal–oxygen bonds corresponding to different oxidation states of the metals [36]. The peak with a binding energy of around 530 eV is related to lattice oxygen in the spinel structure. Peak at 531.9 eV is attributed to defective surface oxide species, while a small contribution at 533.4 eV is assigned to adsorbed water. Different oxidation states of the transition metal ions in the spinel could be beneficial for the catalytic activity.

The Mn 2p spectrum show peaks at 642.5 eV and 654.0 eV, assigned to Mn2p_3/2_ and Mn2p_1/2_, respectively. The Mn2p_3/2_ peak could be fitted with two peaks at 640.4 eV and 641.8 eV, which along with the absence of a satellite at ~647 eV, is related to the simultaneous presence of Mn^2+^ and Mn^3+^ [37].

The Fe2p photoelectron spectra consist of peaks at 711.8 eV and 724.9 eV that are attributed to Fe2p_3/2_ and Fe2p_1/2_, and the satellite at 719.0 eV is 7.2 eV from the Fe2p_3/2_ main line, confirming the presence of Fe^3+^ [38].

The Co2p spectrum presents two features in the Co2p_3/2_ region and two in the Co2p_1/2_ region. The peaks with binding energies at about 780.8 eV and 787.1 eV are assigned to Co2p_3/2_ and its satellite, while the peaks at about 796.3 eV and 803.7 eV are attributed to Co2p_1/2_ and its corresponding satellite. These peaks’ binding energies, along with the spin–orbit splitting of about 15.5 eV [39], could be assigned to Co^3+^ and Co^2+^ species.

It should be noted that after the catalytic experiments, no substantial changes in binding energies and the surface oxidation states of the studied metals are observed, evidencing the structural and surface stability of the prepared spinels.

An interesting insight into the peculiarities of the prepared spinel catalysts is obtained by comparing the bulk and the surface chemical compositions of the samples as derived by EDS and XPS analyses (Table 3).

For the MCF-S sample, the bulk composition is close to the stoichiometric one within the experimental error. However, the surface composition shows manganese enrichment and slight oxygen deficiency. After the catalytic tests, a slight increase in the surface iron content is seen, while the oxygen/metal ratio remains unchanged. For the MCF-CA sample, the bulk contents of Mn and Co are slightly higher than that of Fe. On the other hand, the surface content of Mn and Fe is slightly higher than that of Co, accompanied by similar oxygen deficiency. The surface of the same sample after the catalytic test seems to be enriched in Co at the expense of Fe. The decrease in the oxygen/metal ratio reveals a partial reduction of the metal ions after catalytic reactions. Similar observations of nonstoichiometric surface composition in spinels have been made by other authors [40,41], suggesting predominant exposure of certain crystallographic planes has increased the presence of some cations on the sample surface.

### 3.7. Catalytic Tests

The catalytic activity of the studied spinels, prepared by sucrose and citric acid as fuel in the reaction of complete oxidation of methane, propane, butane, and propane in the presence of water vapors at a stationary state, is shown in Figure 7.

The lowest values for the ”light-off” temperatures (i.e., the temperature for 50% conversion) are achieved in the complete oxidation of butane (T_50_ = 210 °C) on MCF-S and (T_50_ = 230 °C) on MCF-CA. Complete oxidation is reached at 290 °C. The ”light-off” temperatures in methane oxidation are the highest—(T_50_ = 430 °C) on MCF-S and (T_50_ = 465 °C) on MCF-CA. Complete oxidation of methane is not achieved up to 450 °C, but the degree of conversion at this temperature is more than 80%. The ”light-off” temperatures in propane oxidation are between the above-mentioned values (T_50_ = 240 °C) on MCF-S and (T_50_ = 255 °C) on MCF-CA. Complete oxidation is reached at 300 °C. It should be noted the slight decrease in the catalytic activity in the complete oxidation of propane in the presence of water vapor (2 vol.%, i.e., close to the dew point at room temperature), the ”light-off” temperatures are 15–20 °C higher than those in the absence of water vapor. It is worth mentioning that in all catalytic reactions, the spinel, prepared by fuel sucrose, is more active than this prepared by citric acid.

The kinetics parameters for the power-law kinetic model (PWL) are presented in Table 4. The low values for the observed reaction order towards the oxygen (below 0.2) lead to the conclusion about the significant role of the chemisorption, and therefore reaction mechanisms assuming dissociative oxygen adsorption should be further considered.

The reaction orders towards propane have values 0.7–0.8 which indicates weak adsorption on the surface, and mechanisms involving the reaction of propane from the gas phase (as Eley–Rideal) cannot be excluded.

The negative reaction order towards the water vapor shows an inhibition effect, and it is almost one and the same for both MCF-S and MCF-CA samples (ranging at values of −0.1). The close values for the apparent activation energies (E_app_) and the significant differences in the pre-exponential coefficient (k_o_) could be related to a higher number of active sites formed on the catalytic surface of the MCF-S catalyst. The low absolute value of the reaction order towards the water vapor indicates for relatively weak inhibition effect, which is very beneficial for practical applications. The detailed investigation of the reaction mechanism of propane combustion will be the subject of further investigation.

The comparison of the activities of the spinels prepared for this investigation with the data from the literature reveals that the MCF-S catalyst possesses excellent efficiency in the studied oxidation reactions. It shows lower “light-off” temperatures than single Co_3_O_4_ oxide [42], SBA-15-supported binary Co-Mn-oxides [43], and copper, nickel, or manganese ferrites [23,44].

The present study reports for the first time Mn:Co:Fe = 1:1:1 spinel composition obtained by solution combustion approach using sucrose and citric acid as fuel. Both prepared materials comprise intimately interconnected in nanoscale spinel phases, with slight differences in the composition and sizes related to the type of fuel used in the synthesis reaction. The mentioned differences are due to the small variations of the enthalpy of formations of the presented spinel phases rather than to the inhomogeneity. MCF-S shows a higher specific surface area, pore volume, higher amount of small particle fraction, higher thermal stability, and, as a result, more exposed active sites on the sample surface. These features imply the higher catalytic activity of MCF-S in the studied oxidation reactions.

## 4. Conclusions

The solution combustion synthesis using sucrose and citric acid as fuels is an appropriate method for the preparation of triple spinel oxides containing Mn, Co, and Fe in a 1:1:1 ratio between the metals. The prepared spinels are dispersed and thermally stable and do not undergo substantial changes during the catalytic reactions. On the basis of the triple spinel prepared with sucrose, a competitive catalyst for the complete oxidation of hydrocarbons could be developed.

## Figures and Tables

**Figure 1 nanomaterials-12-03900-f001:**
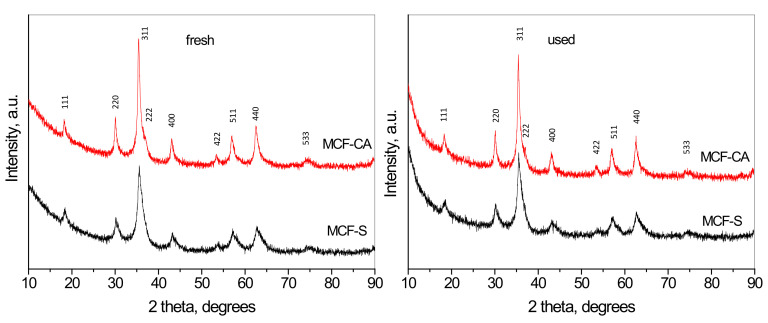
XRD patterns of the fresh and used spinel samples.

**Figure 2 nanomaterials-12-03900-f002:**
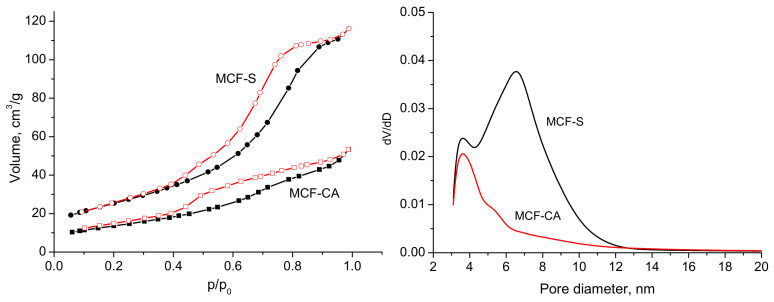
Adsorption–desorption isotherms and pore-size distribution of the spinel ferrite samples.

**Figure 3 nanomaterials-12-03900-f003:**
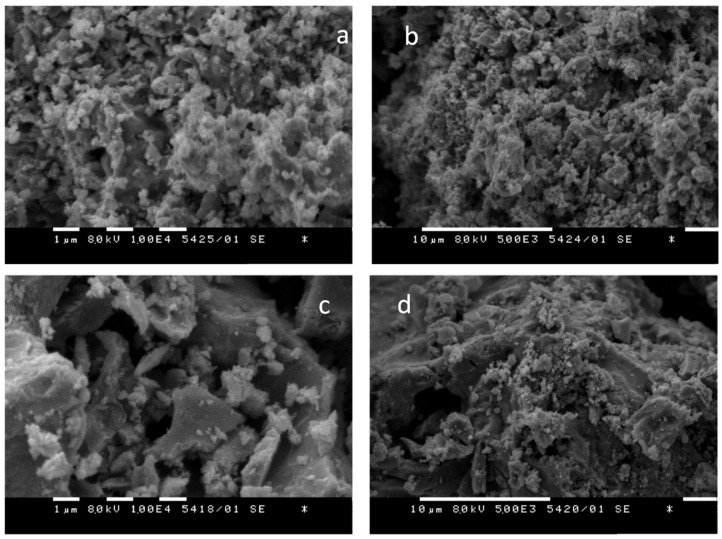
SEM images of MCF-S (**a**,**b**) and MCF-CA (**c**,**d**).

**Figure 4 nanomaterials-12-03900-f004:**
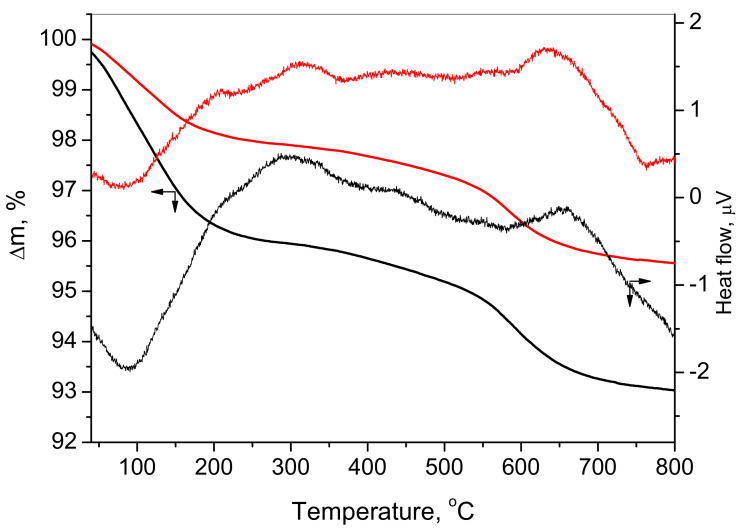
TG and DTA curves of MCF-S (black) and MCF-CA (red).

**Figure 5 nanomaterials-12-03900-f005:**
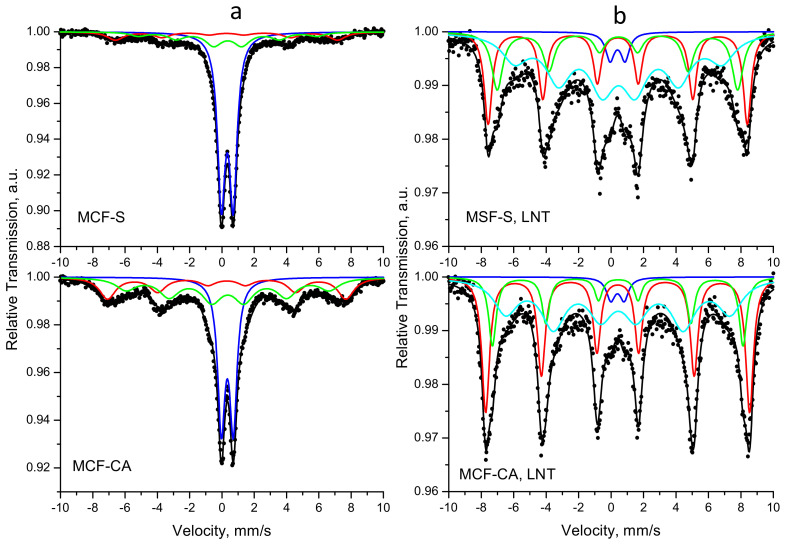
Mössbauer spectra of MCF-S and MCF-CA at 25 °C (**a**) and at −196 °C (**b**).

**Figure 6 nanomaterials-12-03900-f006:**
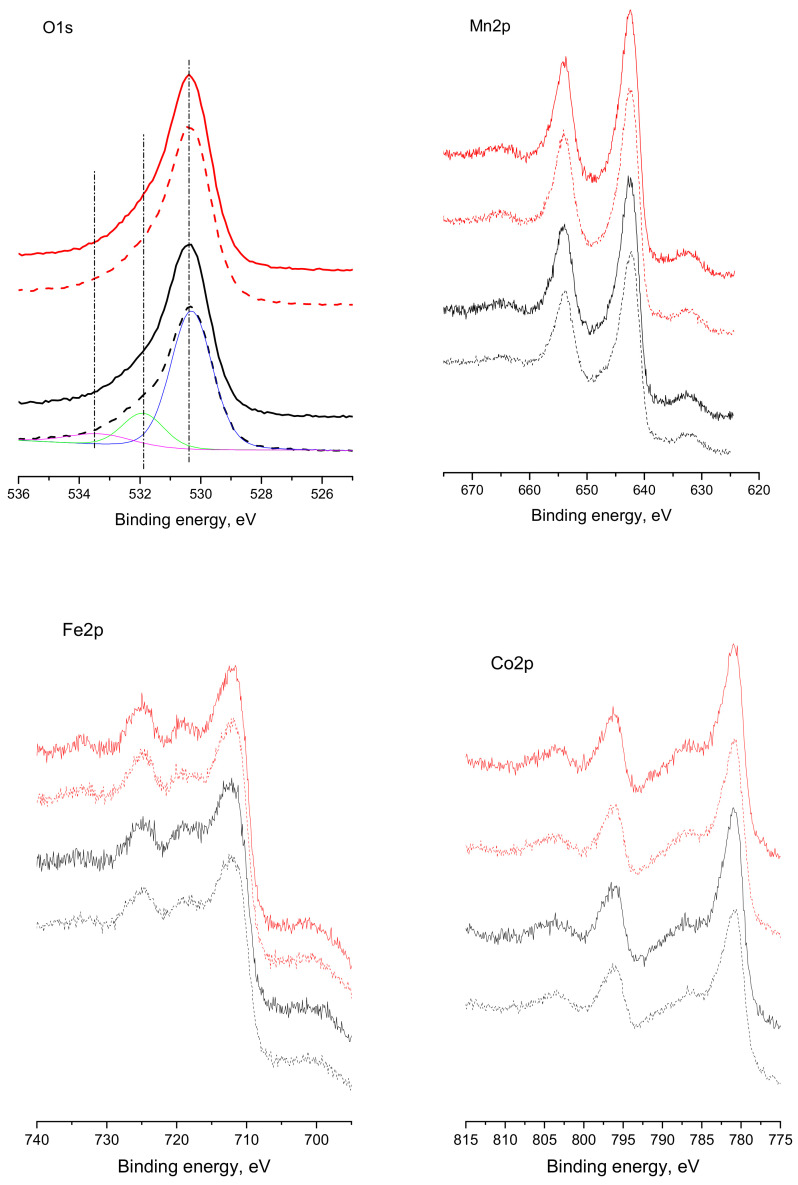
XPS spectra of MCF-S (black) and MCF-CA (red). Dash lines correspond to the same samples after the catalytic reactions. The deconvolution of the MCF-S spectrum after catalytic tests is presented for perspicuity. The same binding energies are observed for the other samples.

**Figure 7 nanomaterials-12-03900-f007:**
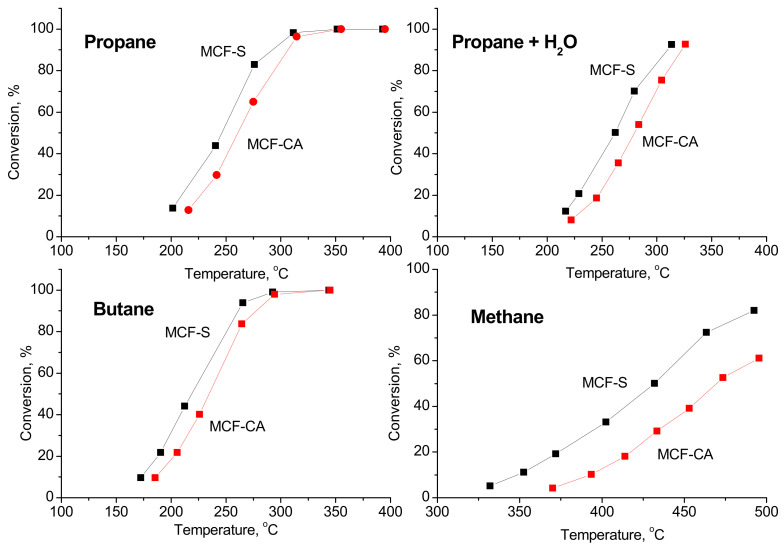
Dependence of the conversion degree on the reaction temperature during the complete oxidation of hydrocarbons on MCF-S and MCF-CA.

**Table 1 nanomaterials-12-03900-t001:** Texture parameters of the synthesized spinel materials.

Sample	S_BET_, m^2^/g	V_t_, cm^3^/g	D_av_, nm
MCF-S	93	0.18	7.8
MCF-CA	50	0.08	6.7

S_BET_—specific surface area; V_t_—total pore volume; D_av_—average pore diameter.

**Table 2 nanomaterials-12-03900-t002:** Mössbauer parameters for MnCoFeO_4_ samples obtained using two types of fuels.

Sample	Components	δ, mm/s	Δ (2ε), mm/s	B_hf_, T	Γ_exp_, mm/s	G, %
MCF-CA	Sx1-Fe^3+^_tetra_	0.30	0.00	45.8	1.22	25
	Sx2-Fe^3+^_octa_	0.36	−0.01	39.0	1.50	37
	Db-Fe^3+^	0.34	0.74	-	0.55	37
MCF-S	Sx1-Fe^3+^_tetra_	0.28	0.00	42.6	1.22	14
	Sx2-Fe^3+^_octa_	0.36	0.00	34.3	1.22	21
	Db-Fe^3+^	0.33	0.74	-	0.57	65
MCF-CA, LNT	Sx1-Fe^3+^_tetra_	0.41	−0.01	50.4	0.57	37
	Sx2-Fe^3+^_octa_	0.43	−0.03	47.9	0.60	16
	Sx3-Fe^3+^	0.44	−0.03	42.9	1.79	44
	Db-Fe^3+^	0.40	0.82	-	0.65	3
MCF-S, LNT	Sx1-Fe^3+^_tetra_	0.41	−0.01	49.5	0.60	27
	Sx2-Fe^3+^_octa_	0.43	−0.07	45.9	0.82	20
	Sx3-Fe^3+^	0.47	−0.02	39.6	1.88	49
	Db-Fe^3+^	0.40	0.92	-	0.65	4

*δ*—isomer shift; Δ (2ε)—quadruple splitting; B_hf_—internal magnetic field; Γ_exp_—the line width; G—relative weight.

**Table 3 nanomaterials-12-03900-t003:** Bulk (EDS) and surface (XPS) chemical compositions of the samples.

Sample	Mn, at. %		Co, at. %		Fe, at. %		O, at. %		O/Me	
	Theor. 14.33		Theor. 14.33		Theor. 14.33		Theor. 57.32		Theor. 1.33	
	EDS	XPS	EDS	XPS	EDS	XPS	EDS	XPS	EDS	XPS
MCF-S	15.1	17.0	15.0	15.5	14.1	15.4	55.8	52.8	1.26	1.10
MCF-S-u	-	16.4	-	14.8	-	16.4	-	52.4	-	1.10
MCF-CA	14.8	16.9	14.1	13.2	13.7	15.8	57.5	54.1	1.35	1.18
MCF-CA-u	-	16.8	-	13.7	-	15.4	-	52.0	-	1.13

Theoretical values are calculated for a spinel composition MnCoFeO_4_.

**Table 4 nanomaterials-12-03900-t004:** Estimated kinetics parameters for the power—law model.

Catalyst	E_app_	k_o_	m (C_3_H_8_)	n (O_2_)	p (H_2_O)	RSS	R^2^
MCF-S	71.3	7.06 × 10^7^	0.77	0.18	−0.12	3.5	0.99
MCF-CA	70.9	8.32 × 10^5^	0.72	0.15	−0.09	9.9	0.95

E_appi_, kJ/mol; k_oi_, mol·s^−1^·m^−3^; k_oi,pwl_, mol·s^−1−[1−(m+n+p)]^; (R^2^)—squared correlation coefficient.

## Data Availability

Not applicable.

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
