# Peer review of "Novel Nanosized Spinel MnCoFeO_4_ for Low-Temperature Hydrocarbon Oxidation"

_nanomaterials, 2022, doi:10.3390/nano12213900_

Round 1

Reviewer 1 Report

The manuscript entitled "Novel nanosized spinel MnCoFeO4 for low temperature hydrocarbons oxidation" by Tumbalev et al. reports the synthesis of spinel MnCoFeO4 catalyst in the presence of glucose and citric acid as reducing agents.The manuscript include many results that are supporting the authors affirmations, but there are still some misunderstandings that make the actual form of the manuscript unpublishable. Further, are listed some recommendations for the manuscript improvement:

- as is stated in the title the nanosized spinel should be "novel", but Elkholy et al. (RSC Adv., 2017, 7, 51888–51895) and Majumdar et al. (Electrochimica Acta, 2021, 385, 138295) already reported such a novel spinel. Thus, in the introduction, the authors should emphasize the novelty of the reported research and highlight their contribution to the field;

- in Table 1 should be SBET instead of S;

- the meaning of each parameter in tables should be written as table footnotes;

- it seems that the MCF-S sample exhibit a dual porosity. Could the authors explain that?

- it is known that catalysts work to speed up the rate of reactions by decreasing the activation energy. Thus, did the authors evaluate the activation energies for the envisaged catalytic reaction and how it is influenced in the presence of the prepared catalysts?

- could the authors comment the catalytic performance of the prepared materials in relation to other catalysts? It will give an added value to the present manuscript.

Author Response

Reviewer 1

We thank the Reviewer for their comments concerning our manuscript. We appreciate the comments and suggestions made by them that we found to be very constructive and helpful. We have studied comments carefully and have made correction, which we hope to meet approval. The changes in the text are marked. The main corrections in the paper and the responds to the Reviewer’s comments are as follows:

The manuscript entitled "Novel nanosized spinel MnCoFeO4 for ow temperature hydrocarbons oxidation" by Tumbalev et al. reports the synthesis of spinel MnCoFeO4 catalyst in the presence of glucose and citric acid as reducing agents.The manuscript include many results that are supporting the authors affirmations, but there are still some misunderstandings that make the actual form of the manuscript unpublishable. Further, are listed some recommendations for the manuscript improvement:

- as is stated in the title the nanosized spinel should be "novel", but Elkholy et al. (RSC Adv., 2017, 7, 51888–51895) and Majumdar et al. (Electrochimica Acta, 2021, 385, 138295) already reported such a novel spinel. Thus, in the introduction, the authors should emphasize the novelty of the reported research and highlight their contribution to the field;

Answer:

The Authors are familiar with the two papers mentioned by the Reviewer. In the paper by Elkholy et al. (RSC Adv., 2017, 7, 51888–51895) the stoichiometric ratio between the Mn, Co, Fe metals is 1:1:2 which differs from that reported in the present manuscript (Mn: Co:Fe = 1:1:1). The composition of the prepared spinel by Majumdar et al. (Electrochimica Acta, 2021, 385, 138295) is somehow ambiguous since they did not mentioned definitely the stoichiometric ratio of metals. We added these two paper to the references mentioning their contribution to the mixed metal spinel synthesis. The two old papers (20,21) contain data for this proper composition, but the synthesis procedure (high temperature ceramic synthesis)  resulted in high crystallinity of the spinels,  moreover the purpose of these works are structural and magnetic behavior of the obtained spinels and no catalytic properties were studied. The novelty of the presented work is in the originality of the composition, the synthesis method allowing nanosized materials with prospective catalytic properties.  

- in Table 1 should be SBET instead of S;

Answer:

The correction was made.

- the meaning of each parameter in tables should be written as table footnotes;

Answer:

The footnote of the Table 1 was added.

- it seems that the MCF-S sample exhibit a dual porosity. Could the authors explain that?
Answer:

Depending on the shape of the particles, the spaces that are formed between them have a very complex shape. When the particles are globular (spherical) the voids formed are similar to toroidal and resemble that which is formed when balls of different sizes are clustered. The shape of these pores is irregular, with different cross-sections and sizes that depend on the number of contacts between them. In the case of plate-like or rod-shaped particles, the shape of the voids between them can be wedge-shaped or slit-shaped, and when the particles are favorable orientated- channels with parallel or almost parallel walls are formed. In our case probably two types of voids are formed between the different shaped particles giving this apparent dual porosity.

- it is known that catalysts work to speed up the rate of reactions by decreasing the activation energy. Thus, did the authors evaluate the activation energies for the envisaged catalytic reaction and how it is influenced in the presence of the prepared catalysts?

Answer:

Within the present study the calculated activation energies are reflection of a complex phenomenon – the reaction rate and the adsorption – desorption enthalpies are temperature dependable and have a cancellation effect to each other. Therefore the text of the manuscript is corrected by the use of the more precise term “apparent” activation energy. Further, the difference in the catalysts behavior could be related to the pre-exponential coefficient ko (called also a “frequency” factor), related with the number of the active sites on the catalytic surface. The text of the manuscript is supplemented accordingly.

- could the authors comment the catalytic performance of the prepared materials in relation to other catalysts? It will give an added value to the present manuscript.

Answer:

A comparison between the performance of the present catalysts and published data for the activity of similar materials as Co3O4 oxide, SBA-15 supported binary Co-Mn-oxides, copper, nickel or manganese ferrites during experiments on the same reaction is added to the manuscript.

Reviewer 2 Report

This work done by Ventsislav Tumbalev et al reports “Novel nanosized spinel MnCoFeO4 for low temperature hydrocarbons oxidation”, showing the MnCoFeO4 spinels prepared by solution combustion method with peculiar composition and their catalytic behavior in the reactions of complete oxidation of hydrocarbons such as methane, propane, butane and propane. The results have been discussed and correlated with the characterization data as well as the catalytic behaviors. This is an interesting work on the subject. Thus, I recommend that this manuscript can be accepted for publication in the journal of Nanomaterials after minor revision. Some issues are listed as follows:

1.       The authors state that MCF-S and MCF-CA all contains two spinels and the crystallite sizes of two spinels have been provided. Please point out two spinels mean what and how to calculate the crystallite sizes of these two spinels.

2.       The MCF-CA owns the macro-mesopores, but its Dav (6.7 nm) is smaller than MCF-S (7.8 nm) that only contains mesopores, why? Please explain.

3.       As the authors state that O1s spectra can be decovoluted into three peaks, but only one peak can be seen from O1s spectra in Figure 6. Please revise it.

4.       Some literatures related to spinel metal oxides and catalytic oxidation of VOCs can be cited to improve the manuscript such as:

Chem. Eng. J. 2018, 337, 488-498,

Catal. Sci. Technol. 2015, 5, 4594-4601,

Reviewer 3 Report

The manuscript (nanomaterials-2009740) described a study on the influence of combustion fuel for the physicochemical properties and the catalytic performance. The topic is interesting, however, some major comments should be addressed. 

Comment 1. Why select triple metal oxide MnCoFeO4 as the study topic, since it seems that the catalytic activity may be not good enough, maybe inferior to that of single Co3O4 catalyst.

Comment 2: It is suggested to go deeper into the analysis of the intrinsic reasons for the difference in the properties and activity. Using sucrose as fuel could provide higher calcination temperature than citric acid? Sucrose containming more carbon content is more reductive compared to that of citric acid?

Comment 3: Please specify the method for the calculation of the mean crystallite size which include two spinels. 

Comment 4: Please show the deconvolution results of the XPS peaks in Fig. 6. Please make comparison of the ratio of the defective surface oxygen, which play important role for the oxidation reaction.

Cooment 5: The grammar and spelling mistakes should be carefully checked, e.g. "patterns were obtained in the range 5-902θ" and other place. Please carefully check the whole manuscript. 

Comment 6: The following relative references are suggested for the citation: Applied Surface Science, 2022, 600, 15404, Fuel, 2022, 314, 122774.

Round 2

Reviewer 1 Report

The manuscript has been improved after revision. The added information is clear presented and increase the quality of the manuscript.

One more recommendation I would give: for each table footnotes should be added.

After that, I consider the manuscript is ready for publication.

Author Response

Thank you for the recommendations. The Tables footnotes were added.

Reviewer 3 Report

The author has well adressed the comments, it is suggested to be accepted without further modifications.

Author Response

Thank you for the recommendations.